# Hi-LASSO: High-performance python and apache spark packages for feature selection with high-dimensional data

**Jongkwon Jo[1], Seungha Jung[1], Joongyang Park[1], Youngsoon Kim[1]\*, Mingon Kang[2]\***

1 Department of Information and Statistics, Gyeongsang National University, Jinju-si, South Korea,
2 Department of Computer Science, University of Nevada, Las Vegas, Nevada, United States of America

\* youngsoonkim@gnu.ac.kr (YK); mingon.kang@unlv.edu (MK)

**Data Availability Statement:** All relevant data are within the paper and its Supporting information files. Hi-LASSO packages are publicly available on GitHub at https://github.com/datax-lab/Hi-LASSO

## Abstract

High-dimensional LASSO (Hi-LASSO) is a powerful feature selection tool for high-dimensional data. Our previous study showed that Hi-LASSO outperformed the other state-of-the-art LASSO methods. However, the substantial cost of bootstrapping and the lack of experiments for a parametric statistical test for feature selection have impeded to apply Hi-LASSO for practical applications. In this paper, the Python package and its Spark library are efficiently designed in a parallel manner for practice with real-world problems, as well as providing the capability of the parametric statistical tests for feature selection on high-dimensional data. We demonstrate Hi-LASSO's outperformance with various intensive experiments in a practical manner. Hi-LASSO will be efficiently and easily performed by using the packages for feature selection. Hi-LASSO packages are publicly available at https://github.com/datax-lab/Hi-LASSO under the MIT license. The packages can be easily installed by Python PIP, and additional documentation is available at https://pypi.org/project/hi-lasso and https://pypi.org/project/Hi-LASSO-spark.

## Introduction

The Least Absolute Shrinkage and Selection Operator (LASSO) and its derivatives have been widely used as powerful linear regression-based feature selection tools that identify a subset of relevant variables in model construction [1]. LASSO's major derivatives include ElasticNet [2], Adaptive LASSO [3], Relaxed LASSO [4], and Precision LASSO [5], as well as bootstrapping-based LASSOs, such as Random LASSO [6] and Recursive Random LASSO [7]. LASSO is a popular feature selection approach for high-dimensional data in various fields, such as Internet of Things, social media, and engineering research [8, 9].

We recently proposed a high-dimensional LASSO (Hi-LASSO) that theoretically improves the predictive power and feature selection performance on High-Dimension, Low Sample Size (HDLSS) data [10]. Hi-LASSO (1) alleviates bias introduced from bootstrapping, (2) satisfies the global oracle property, and (3) provides a Parametric Statistical Test for Feature Selection in Bootstrap regression modeling (PSTFSboot). However, despite of the outstanding

under the MIT license. The packages can be easily installed by Python PIP, and additional documentation is available at https://pypi.org/project/hi-lasso and https://pypi.org/project/Hi-LASSO-spark.

**Funding:** Y.S. Kim is supported for the work by the National Research Foundation of Korea (NRF-2021R1I1A3048029).

**Competing interests:** The authors have declared that no competing interests exist.

performance of Hi-LASSO in feature selection, the substantial cost of bootstrapping and the lack of experiments for a parametric statistical test for feature selection have impeded practical applications of Hi-LASSO.

In this paper, we introduce Hi-LASSO packages implemented in Python and Apache Spark, which improve the efficiency in a parallel manner, as scalable and practical tools for feature selection with HDLSS data. We assessed Hi-LASSO's outstanding performance in feature selection with PSTFSboot, which was not thoroughly explored in the original paper due to the expensive computational cost from extensive bootstrapping. We also provide insights for the optimal hyper-parameter settings from various simulation experiments.

## Materials and methods

Hi-LASSO is a linear regression-based feature selection model that produces outstanding performance in both prediction and feature selection on high-dimensional data, by theoretically improving Random LASSO. We first introduce Random LASSO and its limitations. Then, we present how Hi-LASSO improves the LASSO model in this section.

### Random LASSO and its limitations

Random LASSO introduced bootstrapping for robust analysis with high-dimensional data. Random LASSO consists of two procedures of bootstrapping [6]. The first procedure computes importance scores of predictors while approximating weights of variables by drawing multiple bootstrap samples. The second procedure estimates coefficients of the linear model on bootstrapping samples with weights of the importance scores, where predictors having higher importance scores have higher chances to be selected than lower ones. Then, the final estimations of the coefficients are computed by taking the average of the multiple estimates from the bootstrapping. Random LASSO deals with multicollinearity of different signs and identifies non-zero variables more than the sample size.

However, there are still several issues to improve. First, Random LASSO sets coefficients of predictors to zeros, even though the predictors are never selected in the bootstrapping. The unselected predictors may be possibly estimated as non-zero coefficients if being selected in the bootstrapping. Therefore, it introduces a systematic bias regardless its importance of the predictors. Moreover, the lower number of bootstrapping in Random LASSO generates the more systematic bias, where there may be more variables never selected. Note that the bootstrapping number directly affect computational costs in Random LASSO. Second, Random LASSO does not take advantage of *global* oracle property. Although Random LASSO uses bootstrapping with weights being proportional to importance scores of predictors in the second procedure, the final coefficients are estimated without the weights. Random LASSO may adopt adaptive LASSO to take the oracle property. However, Adaptive LASSO takes *local* weights of each bootstrapping in Random LASSO, where the *local* oracle property may vary depending on which predictors are involved in the bootstrapping. Finally, Random LASSO does not provide a statistical test to identify a set of features from multiple bootstrapping results. Random LASSO considers a heuristic threshold, for example, the reciprocal of the sample size, without statistical test, although the results of the feature selection substantially depend on the threshold.

### Hi-LASSO improves Random LASSO

Hi-LASSO tackles the aforementioned limitations of Random LASSO. The contributions of Hi-LASSO are follows. (1) Hi-LASSO rectifies the systematic bias that Random LASSO introduces, by refining the process to compute importance scores. To prevent the systematic bias,

Hi-LASSO considers the coefficient estimation of the unselected predictors as missing values on the bootstrapping in the procedures. (2) Hi-LASSO computes importance scores of variables by averaging absolute coefficients. The coefficient of a predictor may be assigned a different value or opposite sign with its estimate in different linear models with other predictors, specifically multicollinearity with different signs may often cause coefficient estimates of different signs over bootstrapping. Therefore, taking the absolute value of the sum of the coefficient estimates of bootstrapping in Random LASSO may reduce the importance score. (3) Hi-LASSO provides a statistical strategy to determine the number of bootstrapping. The determination of the number of bootstrap samples is crucial to ensure the performance for high-dimensional data. Some predictors may never be considered due to the nature of random sampling no matter how important they are in the model. Thus, Hi-LASSO considered a sufficient number of bootstrap for all predictors to be taken into account. (4) Hi-LASSO takes advantage of *global* oracle property by adopting Adaptive LASSO [3] in the second procedure. (5) Hi-LASSO uses parametric statistical tests for feature selection in bootstrap regression modeling (PSTFSboot). PSTFSboot allows Hi-LASSO to robustly perform feature selection from multiple bootstrapping results, as a filter feature selection, while most LASSO models are wrapper-based feature selection [9].

## Hi-LASSO packages

We provide efficient solutions for performing Hi-LASSO with high-dimensional data, as Python and Apache Spark packages. We reduce the high-cost computations from a number of independent bootstrapping by using parallel processing. We also improve the scalability of Hi-LASSO by implementing the algorithm based on the Apache Spark engine. The Python package for Hi-LASSO (https://pypi.org/project/hi-lasso) and its Apache Spark version (https://pypi.org/project/hi-lasso-spark) are available through PyPI and can be easily installed using Python PIP:

```
pip install hi-lasso // installation in Python
pip install hi-lasso-spark // installation in Apache Spark (Spark
3.0.0+)
```

Sample codes and troubleshooting guide are provided in the web pages (https://hi-lasso.readthedocs.io/en/latest/). Hi-LASSO includes the following hyper-parameters: '$q_1$', '$q_2$', '$L$', '*alpha*', '*logistic*', '*random_state*', and '*n_jobs*'. In Hi-LASSO, each procedure repeats a random selection of $q_i$ predictors $B$ times, where the subscript $i$ denotes the first or second procedure, $i \in \{1, 2\}$. Smaller $q_i$ causes more bootstrapping, which requires more computation cost. On the other hand, large $q_i$ may make coefficient estimation less accurate, because the random selection of a large number of predictors is likely to include more multicollinearity. $B$ is determined by $L$, which is a desired average number of times that a predictor is selected during the bootstrapping. At the end of the second procedure, the parametric statistical test for feature selection is performed with a threshold *alpha* (e.g., 0.01 or 0.05). *logistic* indicates if the model is based on a logistic regression model for binary classification or a linear regression model for regression problems. *n_jobs* sets the number of cores for parallel processing.

## Results

We conducted intensive experiments to assess the performance of the Hi-LASSO packages with the parametric statistical test (PSTFSboot): (1) We performed a simulation study by generating various dimensional data, where feature selection performance and efficiency were assessed; (2) We assessed Hi-LASSO by using semi-real datasets based on TCGA cancer data

and analyzed hyper-parameter settings and performance in practice; and (3) We further assessed the robustness of Hi-LASSO.

## Simulation study

We assessed the feature selection performance of Hi-LASSO with PSTFSboot by comparing with current state-of-the-art LASSO methods, including LASSO, ElasticNet, Adaptive, Relaxed, Random, Recursive, and Precision. We generated synthetic data with six different scenarios: Dataset I—Dataset VI, where numbers of predictors and samples are varied and the ground truth of relevant features are known (S1 File). We measured F1-scores, where relevant variables ($|\beta| > 0$) were considered as positive, and irrelevant variables ($|\beta| = 0$) as negative in a confusion matrix. F1-scores show how accurately a model identifies the set of relevant features as feature selection [11]. Note that the original paper of Hi-LASSO assessed the feature selection performance using F1-scores by a threshold that maximizes the Root Mean Square Error (RMSE) of the validation data without a parametric statistical test, whereas this study conducted the experiments with further feature selection process that statistically combines bootstrapping results (i.e., using PSTFSboot), which does not require validation data.

We tuned the hyper-parameters of the benchmark models. We optimized the hyper-parameters of Precision and Relaxed LASSO as their original papers proposed. For Hi-LASSO, we set $L$ as 30 and $q_1$ and $q_2$ as the sample size. We set the hyper-parameters of $q$ and $B$ in Random LASSO and Recursive LASSO as Hi-LASSO did for the fair comparison. For the other benchmark models, the optimal hyper-parameters of L1 or L2-norm regularization ($\lambda$) were obtained to minimize the prediction error with inner 5-fold cross validation in the training.

We repeated the experiments ten times by randomly generating simulation data for reproducibility. The experimental results are shown in Table 1 and Fig 1A. Overall, Hi-LASSO outperformed the benchmark models on most of the datasets. Hi-LASSO produced the highest F1-scores in all experiments except Dataset III. However, when sample sizes are increased, Hi-LASSO constantly showed superior performance, at least 10–20% higher F1-scores, for feature selection with high-dimensional data.

We also assessed the efficiency of the Hi-LASSO packages. Fig 1B shows the improved speedup of Hi-LASSO on a parallel processing in Python and Spark, comparing to Hi-LASSO's implementation in the original paper, using a large-scale simulated data (Dataset V) on a machine with Intel Xeon Gold (24 cores × 2). The experimental results show that the packages in Python and Apache Spark enhanced the speedup to 3.75 and 4.83 times faster, respectively. The Spark version was approximately two times faster than the Python version. The details of the execution time are in S2 File.

**Table 1. Experimental results for feature selection performance (F1-scores) with the simulation data.**

| Dataset | LASSO | ElasticNet | Adaptive | Relaxed | Random | Recursive | Precision | Hi-LASSO |
|---|---|---|---|---|---|---|---|---|
| Dataset I (p = 100, n = 50) | 0.6494±0.069 | 0.6837±0.082 | 0.6283±0.080 | 0.5446±0.190 | 0.5856±0.051 | 0.4118±0.084 | 0.6455±0.232 | **0.7256±0.073** |
| Dataset II (p = 100, n = 100) | 0.5582±0.179 | 0.5648±0.179 | 0.6171±0.116 | 0.5302±0.235 | 0.4956±0.063 | 0.4724±0.056 | 0.5457±0.166 | **0.8110±0.034** |
| Dataset III (p = 1,000, n = 100) | 0.1343±0.051 | 0.2730±0.203 | 0.1089±0.038 | 0.1077±0.047 | **0.3092±0.048** | 0.0827±0.038 | 0.2366±0.160 | 0.2697±0.089 |
| Dataset IV (p = 1,000, n = 200) | 0.5236±0.054 | 0.5494±0.073 | 0.4659±0.046 | 0.4944±0.043 | 0.4813±0.033 | 0.1446±0.080 | 0.6289±0.188 | **0.8406±0.037** |
| Dataset V (p = 10,000, n = 200) | 0.3489±0.039 | 0.3660±0.051 | 0.2122±0.063 | 0.2882±0.050 | 0.1657±0.018 | 0.0050±0.011 | 0.4003±0.306 | **0.7117±0.021** |
| Dataset VI (p = 10,000, n = 400) | 0.3224±0.033 | 0.3361±0.030 | 0.2497±0.070 | 0.2958±0.034 | 0.1132±0.004 | 0.0330±0.029 | 0.7320±0.085 | **0.8295±0.039** |

The highest F1-scores are highlighted in bold. $p$ is a number of variables, and $n$ is a sample size.

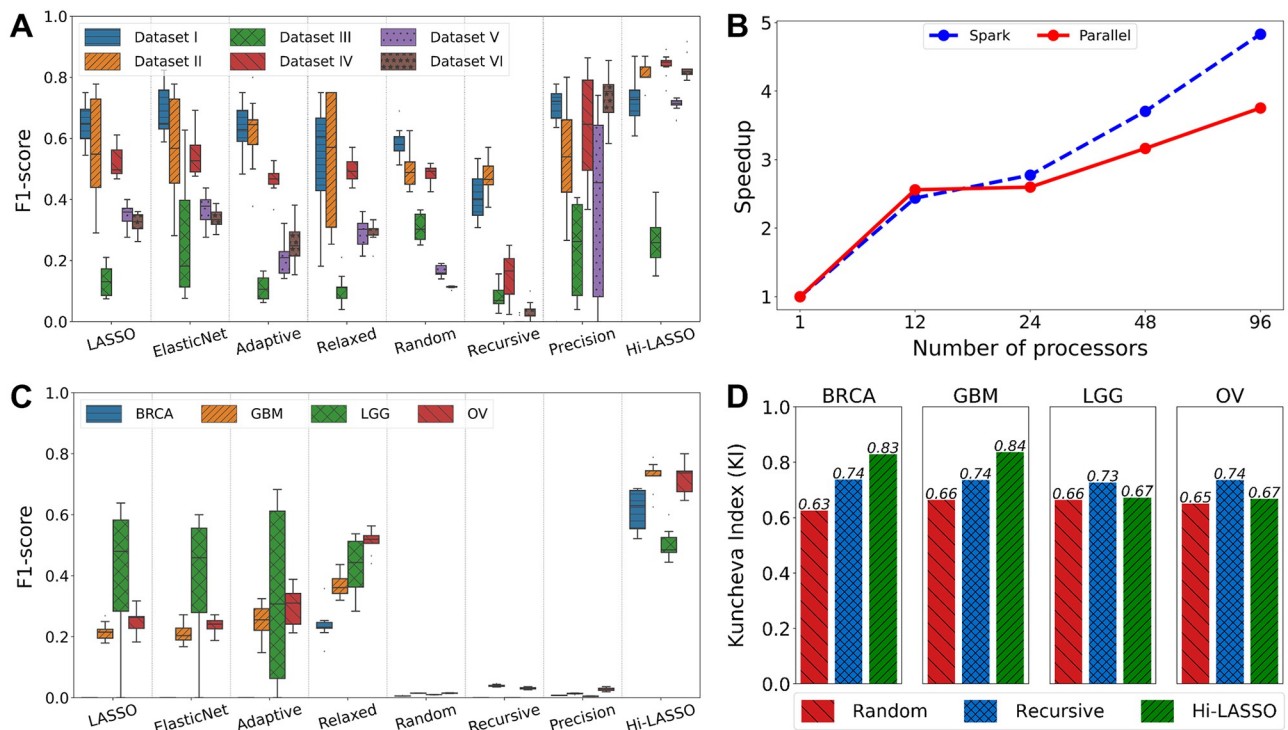

**Fig 1. Experimental results.** (A) Comparison of F1-scores with the simulation data, (B) improvement of efficiency on the Hi-LASSO Python package and the Spark version, (C) comparison of F1-scores with the semi-real simulation data, and (D) robustness of feature selection.

### Semi-real simulation study based on TCGA cancer data

We further conducted a semi-real simulation study based on cancer data, including Glioblastoma Multiforme (GBM), Low Grade Gliomas (LGG), breast cancer (BRCA), and ovarian cancer (OV) in The Cancer Genome Atlas Program (TCGA) repository (https://www.cancer.gov/tcga). We downloaded the cancer genomic data from https://www.cbioportal.org. For the semi-real simulation study, we used gene expression (e.g., microarray or RNA-seq) as independent variables and generated a response variable as follows: (1) We performed correlation analysis between survival months and gene expression; (2) We selected 20 genes with the highest correlation with survival months; (3) The regression coefficients ($\hat{\boldsymbol{\beta}}$) of the 20 genes were randomly generated from the normal distribution, $\mathcal{N}(\mu=4, \sigma^2=1)$, preserving the sign of the correlation coefficient corresponding to the regression coefficient; (4) The coefficients of the other genes were set to 0; and (5) The synthetic survival months (the response variable) were generated from the linear combination of the gene expression ($\mathbf{X}$) and the ground truth's coefficients ($\hat{\boldsymbol{\beta}}$) and errors ($\epsilon$) from the normal distribution with a mean of zero and the standard deviation of the logarithmic survival months, i.e., $\mathbf{y} = \mathbf{X}\hat{\boldsymbol{\beta}} + \epsilon$.

We then calculated F1-scores to evaluate the performance of feature selection with the semi-real data (Table 2 and Fig 1C). The experimental results showed that Random, Recursive, and Precision produced F1-scores close to zeros, whereas LASSO, ElasticNet, and Adaptive showed F1-scores between 0.0 and 0.4. Remarkably, Hi-LASSO presented F1-scores between 0.5 and 0.72. The numbers of non-zero variables identified by the benchmark methods are shown in the parentheses in Table 2. Hi-LASSO constantly identified 10~17 relevant variables, which is closed to the 20 non-zero variables in the ground truth. Whereas, Random and

**Table 2. Experimental results for feature selection performance (F1-scores and numbers of non-zero variables in parentheses) with the semi-simulated data based on four TCGA cancer datasets.** The average of the experiments are shown, where bold-face indicates the best performance.

| Dataset | LASSO | ElasticNet | Adaptive | Relaxed | Random | Recursive | Precision | Hi-LASSO |
|---|---|---|---|---|---|---|---|---|
| BRCA | 0.0000±0.000 | 0.0000±0.000 | 0.0000±0.000 | 0.2380±0.050 | 0.0058±0.001 | 0.0000±0.000 | 0.0078±0.001 | **0.6165±0.067** |
| (p = 20,212, n = 1,099) | (0.1±0.3) | (0.1±0.3) | (1.0±0.0) | (144.9±37.6) | (5286.9±222.9) | (139.5±16.1) | (1689.4±28.0) | **(17.8±3.2)** |
| GBM | 0.2160±0.027 | 0.2107±0.032 | 0.2515±0.057 | 0.3695±0.038 | 0.0151±0.000 | 0.0395±0.003 | 0.0134±0.001 | **0.7231±0.046** |
| (p = 12,042, n = 524) | (167.8±22.9) | (173.8±28.9) | (143.0±43.5) | (88.8±11.8) | (2445.8±35.5) | (82.0±8.5) | (1825.2±216.9) | **(12.6±1.1)** |
| LGG | 0.3995±0.242 | 0.3815±0.230 | 0.3412±0.281 | 0.4341±0.088 | 0.0101±0.001 | 0.0000±0.000 | 0.0050±0.001 | **0.5013±0.047** |
| (p = 20,167, n = 529) | (34.5±29.2) | (41.3±28.2) | (43.2±49.8) | (68.8±22.4) | (3221.0±75.8) | (80.6±10.5) | (2893.0±626.5) | **(13.6±2.3)** |
| OV | 0.2530±0.038 | 0.2389±0.024 | 0.2985±0.064 | 0.5120±0.036 | 0.0151±0.001 | 0.0311±0.003 | 0.0273±0.005 | **0.7213±0.054** |
| (p = 12,042, n = 532) | (141.6±26.3) | (149.2±19.0) | (118.5±30.7) | (58.5±5.9) | (2554.9±79.4) | (109.9±14.4) | (1491.0±303.9) | **(16.3±2.4)** |

Precision tended to select large numbers of variables (>1,000 non-zeros), which caused recall relatively higher but precision closed to zero. It may be because Random LASSO uses a reciprocal of sample size as a threshold for feature selection, and Precision LASSO is optimized to regression problems rather than feature selection. Recursive LASSO tends to introduce extreme bias in the first bootstrapping for feature selection, which makes it often fail to identify relevant features. LASSO, ElasticNet, Adaptive, and Relaxed showed unstable performance with large variance on the numbers of non-zero variables in the semi-real data.

## Tuning the hyper-parameters

The optimization of the hyper-parameters, $q_1$, $q_2$, and $L$, is often critical to the performance of feature selection in Hi-LASSO. We investigated how the hyper-parameters affect the performance of Hi-LASSO using the two simulation data (S3 File). We compared F1-scores by varying values of the hyper-parameters, where we set the identical values for $q_1$ and $q_2$ (i.e., $q = q_1 = q_2$) for the sake of simplicity. We empirically found that the optimal values of $q$ were around of the sample size. Generally, the larger $L$ improved the performance in the experiments. However, $L > 50$ does not improve the performance significantly in the experiments. Empirically, the optimal value of $L$ was 30, which approximates normal distribution by the central limit theorem, when the distribution is unknown.

## Robustness for feature selection

Finally, we evaluated the Hi-LASSO's robustness on the feature selection by calculating pairwise Kuncheva Index (KI) on the semi-real simulation datasets [12]. Specifically, Hi-LASSO, Random, and Recursive LASSO are based on bootstrapping, which may produce different results on every execution. KI calculates how much two sets are overlapped, where one indicates two sets are identical, and zero shows no overlap between the two sets. Hi-LASSO showed KI of 0.751 on average on the four cancer semi-real data, whereas Random and Recursive showed 0.650 and 0.733, respectively (Fig 1D). Note that Hi-LASSO produced the highest F1-scores on the semi-real simulation data. The best F1-scores and robustness with the highest KIs demonstrated reliable feature selection performance of Hi-LASSO.

## Conclusion

We introduce Hi-LASSO packages in Python and Apache Spark, which improve the efficiency and scalability for feature selection with high-dimensional data. The Hi-LASSO packages can be easily installed by Python PIPs and can efficiently and effectively analyze high-dimensional data. We demonstrated the extraordinary performance of feature selection with a parametric

statistical test through intensive simulation studies and provide insight how to tune the hyper-parameters in this paper. Hi-LASSO is a promising feature selection tool applicable to practice on real world data.

## Supporting information

**S1 File. Simulation study.**
(PDF)

**S2 File. Performance for efficiency.**
(PDF)

**S3 File. Tuning the hyper-parameters in Hi-LASSO.**
(PDF)

**S4 File. Robustness analysis using Kuncheva Index (KI).**
(PDF)

## Author Contributions

**Conceptualization:** Mingon Kang.

**Investigation:** Jongkwon Jo, Joongyang Park, Youngsoon Kim, Mingon Kang.

**Methodology:** Jongkwon Jo, Seungha Jung, Youngsoon Kim, Mingon Kang.

**Project administration:** Mingon Kang.

**Software:** Jongkwon Jo, Seungha Jung.

**Supervision:** Joongyang Park, Youngsoon Kim, Mingon Kang.

**Visualization:** Jongkwon Jo, Seungha Jung.

**Writing – original draft:** Jongkwon Jo.

**Writing – review & editing:** Youngsoon Kim, Mingon Kang.

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
