## [Decision Letter · Decision Letter 0]

28 Jul 2022

PONE-D-22-19015Hi-LASSO: High-performance Python and Apache spark packages for feature selection with high-dimensional dataPLOS ONE

Dear Dr. Kang,

Thank you for submitting your manuscript to PLOS ONE. After careful consideration, we feel that it has merit but does not fully meet PLOS ONE’s publication criteria as it currently stands. Therefore, we invite you to submit a revised version of the manuscript that addresses the points raised during the review process.

We look forward to receiving your revised manuscript.

Kind regards,

Sathishkumar V E

Academic Editor

PLOS ONE

Journal Requirements:

"This research was supported by the National Research Foundation of Korea (NRF-2021R1I1A3048029). "

"Y.S. Kim is supported for the work by the National Research Foundation of Korea (NRF-2021R1I1A3048029)."

Reviewers' comments:

Reviewer's Responses to Questions

**Comments to the Author**

1. Is the manuscript technically sound, and do the data support the conclusions?

Reviewer #1: No

Reviewer #2: Yes

Reviewer #3: Yes

2. Has the statistical analysis been performed appropriately and rigorously? 

Reviewer #1: No

Reviewer #2: Yes

Reviewer #3: Yes

3. Have the authors made all data underlying the findings in their manuscript fully available?

Reviewer #1: No

Reviewer #2: Yes

Reviewer #3: No

4. Is the manuscript presented in an intelligible fashion and written in standard English?

Reviewer #1: No

Reviewer #2: Yes

Reviewer #3: Yes

5. Review Comments to the Author

Reviewer #1: The Research Paper stands Rejected and is NOT RECOMMENDED for Publication because of the following strong reasons:

1. The overall presentation and conceptual methodology of the paper is very weak and lots of advanced papers are already published.

2. No Strong analysis and experimental results are observed in the paper.

3. No Novelty is there.

4. It is the work of simple theoretical description but even the actual research orientation is missing in the paper.

Reviewer #2: Few experiments can be repeated or justified for f1 scores. The literature study can strengthen with more recent papers. The authors can state how the current standards are maintained, materials and methods are not cited with previous works. the authors can consider the below works for better literature

-Y. Lu, L. Yang, S. X. Yang, Q. Hua, A. K. Sangaiah, T. Guo, and K. Yu, “An Intelligent Deterministic Scheduling Method for Ultra-Low Latency Communication in Edge Enabled Industrial Internet of Things,” IEEE Transactions on Industrial Informatics, 2022, doi: 10.1109/TII.2022.3186891.

J. Wei, Q. Zhu, Q. Li, L. Nie, Z. Shen, K. -K. R. Choo, K. Yu, “A Redactable Blockchain Framework for Secure Federated Learning in Industrial Internet-of-Things”, IEEE Internet of Things Journal, doi: 10.1109/JIOT.2022.3162499.

-Subbiah, S.S. and Chinnappan, J., 2021. Opportunities and Challenges of Feature Selection Methods for High Dimensional Data: A Review. Ingénierie des Systèmes d'Information, 26(1).

-Bolón-Canedo, V. and Alonso-Betanzos, A., 2019. Ensembles for feature selection: A review and future trends. Information Fusion, 52, pp.1-12.

-Y. He, L. Nie, T. Guo, K. Kaur, M. M. Hassan, and K. Yu," A NOMA-Enabled Framework for Relay Deployment and Network Optimization in Double-Layer Airborne Access VANETs," IEEE Transactions on Intelligent Transportation Systems, doi: 10.1109/TITS.2021.3139888.

Reviewer #3: This paper presents a new implementation of a previously published algorithm called Hi-LASSO, with parallel computations that make the algorithm more practical for use with large real-world data sets. It shows experiments on both synthetic and real data that demonstrate the algorithm's utility for feature selection in high-dimensional data sets. Comparisons to a Spark implementation are shown, with performance results indicating the scalability of the method. Finally, the work describes the model's hyperparameters and robustness. I found the paper to be relatively well written overall, with a reasonable order of its sections. This paper will be a good candidate for PLOS ONE with some or all of the revisions suggested below.

I have broken my comments into a few sections focused on the paper, figures, grammar, reproducibility, references, and the code described in the paper.

Paper comments:

- The previous paper on this algorithm used Relative Model Error, Root Mean Square Error, and F1 scores. Why are only F1 scores reported in this work? An explanation of the choice of metric would help strengthen the data.

- The hyperparameters q1, q2, L, alpha should be described in further detail. These are described a little bit in the "tuning" section. However, it would be helpful to know not only the trends in how performance is affected, but also how to choose an initial value for each. It appears there is an "auto" setting in the Python package but that automatic behavior is not described in the paper from what I could tell.

- Were hyperparameters optimized for all LASSO algorithms? How did the authors ensure that all algorithms were fairly assessed? It is surprising to see so many algorithms with F1 scores of zero in the BRCA dataset. Similarly, it is surprising to see the results in Table S5. Is there another dataset that shows a nonzero score for some of the compared algorithms?

- I find it a little hard to believe that Hi-LASSO is this much better than similar algorithms without more information about how each algorithm was run, to ensure fairness in the assessment. Are there cases where Hi-LASSO performs poorly? If so, it would be helpful to include such a case for a baseline. How does Hi-LASSO perform in lower-dimensional cases with more data where other LASSO algorithms have been used in the past? Comparisons like this would help reduce the sense that the datasets are cherry-picked for Hi-LASSO's benefit, and would help to illuminate the contrast between prior art and this algorithm's improvements for specific types of problems.

- Some of the results are a bit surprising, with several comparison methods yielding few or no positive results. This may indicate the selection of overly specific benchmark data sets, or a lack of competitive algorithms for comparison. A bit more explanation of the results in these areas would benefit the reader as well as make the work more defensible. The authors' claim of "extraordinary performance" appears to be somewhat supported by the data that is presented, but it is a little unclear whether this is due to a selective choice of benchmarks. Understanding where the algorithm fails (or performs in an "average" way) is important for readers who wish to make practical use of the package.

- The introduction or conclusions should spend more time contextualizing this algorithm. What fields should consider adopting Hi-LASSO? Genomics may be one such candidate, but other potential applications should be described.

- It would be good to summarize the contributions of each author to the work, perhaps using a standardized framework like CRediT (Contributor Roles Taxonomy).

Figure comments:

- Figure 1 is hard to read and should be higher resolution - ideally a vector graphic format like PDF or EPS. Same for supplementary figures S1, S2.

- Figure 1(B) could be replaced by a scatter plot showing weak scaling performance for the process parallel and Spark implementations from 1 core to the number of cores in the benchmarking machine. This could be for one dataset, or a geometric average of a few datasets. Weak scaling plots are far more useful to understand computational efficiency than a raw speedup chart with no clear baseline. It's not clear if the speedup is linear with the number of cores, which a weak scaling plot would help indicate.

Grammatical / typographical comments:

- Line 24: "impeded to apply Hi-LASSO for practical applications" should say "impeded practical applications of Hi-LASSO"

- Line 111: should say "desired average number of times"

- Line 156: "the Apache version" should say "the Apache Spark version." Apache Spark (or Spark for short) is the proper name of the library -- not just "Apache."

- Line 198: missing a subscript on q1

References / reproducibility comments:

- The TCGA data sets should be cited. https://www.cancer.gov/about-nci/organization/ccg/research/structural-genomics/tcga/using-tcga/citing-tcga

- In accordance with the PLOS ONE "Exceptions to sharing materials" (https://journals.plos.org/plosone/s/materials-software-and-code-sharing), the "authors should include a statement in their Materials and Methods discussing any restrictions on availability or use." It appears the TCGA data is subject to controlled access. This should be made clear to the reader, with information about how to access these controlled datasets (if possible) in order to make the results reproducible.

- The code used to generate synthetic Datasets I - IV does not appear to be included in the linked GitHub repository (I looked in the benchmark models and sample data directories). That should be included to meet PLOS ONE data sharing policies, along with a script to execute the code in the benchmark models directory for all benchmarks on the synthetic data.

- Check the capitalization of journal names and article titles in the references section. Some have unexpected lowercase letters.

- Please cite all relevant scientific software packages used in the hi_lasso software, such as NumPy and SciPy. See https://numpy.org/citing-numpy/ and https://scipy.org/citing-scipy/ for examples.

Code comments:

- Line 116 of the paper: Rather than describing both "parallel" and "n_jobs", just let "n_jobs" default to 1 (the serial case). Then only one parameter is needed, and "parallel" can be removed. A special value of "n_jobs is None" or "n_jobs == 0" could use the number of CPU cores returned by "multiprocessing.cpu_count()" for automatic parallelization across all available cores.

- The choice of the MIT license is good for future works to build on this one!

- Could the Spark and non-Spark libraries be combined, or make the Spark library use the base Python library as a dependency? The two code paths look fairly unrelated right now.

- The "simulation_data" folder on GitHub could include a README that indicates where the data came from or how it was generated.

It is my hope that the authors will consider adapting this algorithm for inclusion in a popular toolkit such as scikit-learn after publication. It seems like a helpful algorithm.

6. PLOS authors have the option to publish the peer review history of their article (what does this mean?). If published, this will include your full peer review and any attached files.

Reviewer #1: No

Reviewer #2: No

Reviewer #3: No

---

## [Author Response · Author response to Decision Letter 0]

17 Nov 2022

We attached the response letter, but the below is a text version:

Editor

[Concern #1] Please ensure that your manuscript meets PLOS ONE's style requirements, including those for file naming.

Response: We carefully checked PLOS ONE’s style requirements, including file naming. We separately uploaded figures and supplementary documents. 

[Concern #2] We note that the grant information you provided in the ‘Funding Information’ and ‘Financial Disclosure’ sections do not match.

Response: We correctly updated the grant information in the submission system. 

[Concern #3] Thank you for stating the following in the Acknowledgments Section of your manuscript: 

"This research was supported by the National Research Foundation of Korea (NRF-2021R1I1A3048029). "

We note that you have provided funding information that is not currently declared in your Funding Statement. However, funding information should not appear in the Acknowledgments section or other areas of your manuscript. We will only publish funding information present in the Funding Statement section of the online submission form. Please remove any funding-related text from the manuscript and let us know how you would like to update your Funding Statement. Currently, your Funding Statement reads as follows: 

"Y.S. Kim is supported for the work by the National Research Foundation of Korea (NRF-2021R1I1A3048029)."

Response: We appreciate it. We removed the funding sentence in the acknowledgments and add it in the online submission form.

[Concern #4] Please include captions for your Supporting Information files at the end of your manuscript, and update any in-text citations to match accordingly.

Response: We added the supporting information at the end of the manuscript, as requested. 

We moved several sections from the supplementary to the main manuscript for “package installation” in Page 3, 

[Concern #5] Please review your reference list to ensure that it is complete and correct.

Response: We carefully checked the reference and updated all incorrect ones. 

Reviewer # 1

[Concern #1] The overall presentation and conceptual methodology of the paper is very weak and lots of advanced papers are already published. No Strong analysis and experimental results are observed in the paper. No Novelty is there. It is the work of simple theoretical description but even the actual research orientation is missing in the paper.

Response: We apologize for the confusion. The manuscript is “application note” rather than “original research”, which introduces useful Python and Apache Spark libraries of Hi-LASSO for high-dimensional feature selection. However, the manuscript demonstrates an innovative concept of Hi-LASSO with statistical significance test with efficient implementation of bootstrapping in a parallel manner, and we conducted intensive experiments showing that the Python and Apache Spark libraries produce outstanding feature selection performance comparing to current benchmark LASSO models. 

The original paper demonstrated the capability of LASSO as both a regression model and a feature selection approach. Whereas, this paper mainly improved parametric statistical tests for feature selection on high-dimensional data. Moreover, the original paper of Hi-LASSO assessed the feature selection performance using F1-scores by a threshold that maximizes the Root Mean Square Error (RMSE) of the validation data without a parametric statistical test, whereas this study conducted the experiments with further feature selection process that statistically combines bootstrapping results (i.e., using PSTFSboot), which does not require validation data. Furthermore, we introduce practical settings for tuning of hyperparameters in Hi-LASSO with various experiments. We also showed robustness of Hi-LASSO in the experiments. Therefore, we believe that this application note would be impactful and valuable as a general feature selection tool for a number of applications. 

 

Reviewer # 2

[Concern #1] Few experiments can be repeated or justified for f1 scores. The literature study can strengthen with more recent papers. The authors can state how the current standards are maintained, materials and methods are not cited with previous works. the authors can consider the below works for better literature

Response: We appreciate the constructive comment. Feature selection can use various evaluation metrics. Since we used simulation data where ground truths are known, we used F1-scores which is a balanced measurement between precision and recall. F1-scores show how accurately Hi-LASSO can select true features in the models. We considered relevant variables (|B|>0) as positive and irrelevant variables (b=0) as negative in a confusion matrix. We cited the literature [11], as the reviewer suggested in Page 4.

We also added the sentence below to clarify Hi-LASSO’s advantage in Page 3 using the reference [9] that the reviewer suggested, as below: 

“PSTFSboot allows Hi-LASSO to robustly perform feature selection from multiple bootstrapping results, as a filter feature selection, while most LASSO models are wrapper-based feature selection [9]”

The original paper of Hi-LASSO assessed the feature selection performance using F1-scores by a threshold that maximizes the Root Mean Square Error (RMSE) of the validation data without a parametric statistical test, whereas this study conducted the experiments with further feature selection process that statistically combines bootstrapping results (i.e., using PSTFSboot), which does not require validation data.

[9] Subbiah, Siva Sankari, and Jayakumar Chinnappan. Opportunities and Challenges of Feature Selection Methods for High Dimensional Data: A Review. Ing ´enierie des Syst`emes d’Information 26.1 (2021).

[11] Bol ´on-Canedo, Ver ´onica, and Amparo Alonso-Betanzos. Ensembles for feature selection: A review and future trends. Information Fusion 52 (2019): 1-12.

 

Reviewer # 3

[Concern #1] This paper presents a new implementation of a previously published algorithm called Hi-LASSO, with parallel computations that make the algorithm more practical for use with large real-world data sets. It shows experiments on both synthetic and real data that demonstrate the algorithm's utility for feature selection in high-dimensional data sets. Comparisons to a Spark implementation are shown, with performance results indicating the scalability of the method. Finally, the work describes the model's hyperparameters and robustness. I found the paper to be relatively well written overall, with a reasonable order of its sections. This paper will be a good candidate for PLOS ONE with some or all of the revisions suggested below.

Response: We appreciate the nice summary of the contribution in the study. 

[Concern #2] The previous paper on this algorithm used Relative Model Error, Root Mean Square Error, and F1 scores. Why are only F1 scores reported in this work? An explanation of the choice of metric would help strengthen the data.

Response: We appreciate the constructive comment and apologize for the confusion. The original paper demonstrated the capability of LASSO as both a regression model and a feature selection approach. Whereas, this paper mainly improved parametric statistical tests for feature selection on high-dimensional data. F1-scores show how accurately Hi-LASSO can select true features in the models. Note that the original paper of Hi-LASSO assessed the feature selection performance using F1-scores by a threshold that maximizes the Root Mean Square Error (RMSE) of the validation data without a parametric statistical test, whereas this study conducted the experiments with further feature selection process that statistically combines bootstrapping results (i.e., using PSTFSboot), which does not require validation data. Meanwhile, Lower Relative Model Error and Root Mean Square Errors are already proven in the previous paper. We updated the justification for the choice of the metric in Page 4 as below: 

“Note that the original paper of Hi-LASSO assessed the feature selection performance using F1-scores by a threshold that maximizes the Root Mean Square Error (RMSE) of the validation data without a parametric statistical test, whereas this study conducted the experiments with further feature selection process that statistically combines bootstrapping results (i.e., using PSTFSboot), which does not require validation data.”

[Concern #3] The hyperparameters q1, q2, L, alpha should be described in further detail. These are described a little bit in the "tuning" section. However, it would be helpful to know not only the trends in how performance is affected, but also how to choose an initial value for each. It appears there is an "auto" setting in the Python package but that automatic behavior is not described in the paper from what I could tell. 

Response: We apologize for the confusion. “Tuning of hyper-parameters” was described in the supplementary in detail, but now we added the section in the main manuscript in Page 6, because it is important information for parameter tuning. In the section, we provide not only the trends in how performance is affected, but also how to choose an initial value as default. Also, we changed “auto” to “default” setting in the python package to avoid the confusion. We described how we chose the default values in the section of “tuning of hyper-parameters”, as below:

“The optimization of the hyper-parameters, q_1, q_2, and L, is often critical to the performance of feature selection in Hi-LASSO. We investigated how the hyper-parameters affect the performance of Hi-LASSO using the two simulation data (S3_File). We compared F1-scores by varying values of the hyper-parameters, where we set the identical values for q_1 and q_2 (i.e., q=q_1=q_2) for the sake of simplicity. We empirically found that the optimal values of q were around of the sample size. Generally, the larger L improved the performance in the experiments. However, L > 50 does not improve the performance significantly in the experiments. Empirically, the optimal value of L was 30, which approximates normal distribution by the central limit theorem, when the distribution is unknown.”

[Concern #4] Were hyperparameters optimized for all LASSO algorithms? How did the authors ensure that all algorithms were fairly assessed? It is surprising to see so many algorithms with F1 scores of zero in the BRCA dataset. Similarly, it is surprising to see the results in Table S5. Is there another dataset that shows a nonzero score for some of the compared algorithms?

Response: We apologize for the confusion. We followed the hyper-parameter optimization strategies for Precision and Relaxed LASSO, as their original papers proposed. For Hi-LASSO, we set L as 30 and q1 and q2 as the sample size, according to the experiments of hyper-parameter tuning (We moved the section from the supplementary to the main manuscript). Random LASSO and Recursive Random LASSO do not provide instructions for hyper-parameters, so we set the same hyper-parameters as Hi-LASSO. For the other benchmark models, the optimal hyper-parameters of L1 or L2-norm regularization \\mathbit{\\lambda} were obtained to minimize the prediction error with inner 5-fold cross validation in the training data. We repeated the experiments ten times by randomly generating simulation data for reproducibility. We updated the information in the manuscript in Page 4.

For the issue of zero F1-scores in BRCA, many LASSOs showed unstable results in feature selection, where most conventional LASSO seldom identified relevant features and Random/Precision identified too many non-zero features. Note that the datasets are semi-real data, where ground truth non-zero variables were known. It may be because of hyper-parameter of “lambda”, which optimized with inner cross validation in the training data. The benchmark models performed relatively well on the other datasets of GBM, LGG, and OV. So, we don’t believe that the implementation was incorrect. Whereas, Hi-LASSO with PSTFSboot performed stably for the feature selection on the all datasets. Table S5 was mistakenly included in the supplementary, which is not relevant experimental results. 

[Concern #5] I find it a little hard to believe that Hi-LASSO is this much better than similar algorithms without more information about how each algorithm was run, to ensure fairness in the assessment. Are there cases where Hi-LASSO performs poorly? If so, it would be helpful to include such a case for a baseline. How does Hi-LASSO perform in lower-dimensional cases with more data where other LASSO algorithms have been used in the past? Comparisons like this would help reduce the sense that the datasets are cherry-picked for Hi-LASSO's benefit, and would help to illuminate the contrast between prior art and this algorithm's improvements for specific types of problems.

Response: We apologize for the confusion. We believe that our implementation and experiments were fair to all the benchmark models, as we elucidated the experiment settings and repeated the simulation experiments multiple times. We also clarified how we tuned the hyper-parameters on each model. Although Hi-LASSO with PSTFSboot showed the best performance in the most experiments, there was a case that Random LASSO was better in Table 1. 

Hi-LASSO is mainly designed for High-Dimensional, but Low-Sample Size (HDLSS) data. So, we did not consider lower-dimensional data in the experiments (Lower-dimensional data is not our scope of the study). However, we expect that Hi-LASSO is a still useful model for the lower-dimensional data because we provide statistical test for feature selection using PSTFSboot. 

We did not cherry-picked datasets. The simulation setting is very widely-used for the simulation study in LASSO, where we considered very variety settings for high-dimensional, low sample size data. 

We consider the four semi synthetic data using TCGA datasets, because the four datasets are with very large sample sizes (also very high-dimensional) comparing to the other cancer datasets. 

[Concern #6] Some of the results are a bit surprising, with several comparison methods yielding few or no positive results. This may indicate the selection of overly specific benchmark data sets, or a lack of competitive algorithms for comparison. A bit more explanation of the results in these areas would benefit the reader as well as make the work more defensible. The authors' claim of "extraordinary performance" appears to be somewhat supported by the data that is presented, but it is a little unclear whether this is due to a selective choice of benchmarks. Understanding where the algorithm fails (or performs in an "average" way) is important for readers who wish to make practical use of the package.

Response: We apologize for the confusion. We considered simulated data and semi-real datasets for the assessment. The experimental settings are conventional for the simulation studies, so we do not believe that the experiment datasets are flavored to our model. 

We considered seven benchmark LASSO models, including conventional LASSO (1996), ElasticNet (2005), Adaptive (2006), Relaxed (2007), Random (2011), Recursive (2015), and Precision (2019), which are representatives of LASSO variations. 

[Concern #7] The introduction or conclusions should spend more time contextualizing this algorithm. What fields should consider adopting Hi-LASSO? Genomics may be one such candidate, but other potential applications should be described.

Response: We appreciate the constructive comment. Applications in any fields involving high-dimensional data can leverage Hi-LASSO. We added it in the main manuscript in Page 2 as below: 

“LASSO is a popular feature selection approach for high-dimensional data in various fields, such as biomedical, Internet of Things, social media, and engineering research [8, 9]”

[8] Wang, Chen, and Liu. Establish algebraic data-driven constitutive models for elastic solids with a tensorial sparse symbolic regression method and a hybrid feature selection technique. Journal of the mechanics and physics of Solid, 2022.

[9] Subbiah, Siva Sankari, and Jayakumar Chinnappan. Opportunities and Challenges of Feature Selection Methods for High Dimensional Data: A Review. Ing ´enierie des Syst`emes d’Information 26.1 (2021).

[Concern #8] It would be good to summarize the contributions of each author to the work, perhaps using a standardized framework like CRediT (Contributor Roles Taxonomy).

Response: We added the section of “Author contributions” in Page 7. 

[Concern #9] Figure 1 is hard to read and should be higher resolution - ideally a vector graphic format like PDF or EPS. Same for supplementary figures S1, S2.

Response: We appreciate the constructive comment. We updated the figure to clearly show the experimental results in Page 5. The figure is now with 300 dpi. 

[Concern #10] Figure 1(B) could be replaced by a scatter plot showing weak scaling performance for the process parallel and Spark implementations from 1 core to the number of cores in the benchmarking machine. This could be for one dataset, or a geometric average of a few datasets. Weak scaling plots are far more useful to understand computational efficiency than a raw speedup chart with no clear baseline. It's not clear if the speedup is linear with the number of cores, which a weak scaling plot would help indicate.

Response: We appreciate the constructive comment. We updated the figure 1b with speedup over various numbers of processors (1-96) and compared the efficiency between the spark and python version with the baseline of the initial paper of a single process. 

[Concern #11] Grammatical / typographical comments:

- Line 24: "impeded to apply Hi-LASSO for practical applications" should say "impeded practical applications of Hi-LASSO"

- Line 111: should say "desired average number of times"

- Line 156: "the Apache version" should say "the Apache Spark version." Apache Spark (or Spark for short) is the proper name of the library -- not just "Apache."

- Line 198: missing a subscript on q1

Response: We appreciate. We corrected the grammar errors.

[Concern #12] References / reproducibility comments:

- The TCGA data sets should be cited. 

Response: We appreciate. We added the link in the footnote for the dataset. 

[Concern #13]

- In accordance with the PLOS ONE "Exceptions to sharing materials" (https://journals.plos.org/plosone/s/materials-software-and-code-sharing), the "authors should include a statement in their Materials and Methods discussing any restrictions on availability or use." It appears the TCGA data is subject to controlled access. This should be made clear to the reader, with information about how to access these controlled datasets (if possible) in order to make the results reproducible. 

Response: We appreciate the comments. We downloaded the cancer genomic data from https://www.cbioportal.org, which does not have access restrictions. We added it in Page 5 as below: 

“We downloaded the cancer genomic data from https://www.cbioportal.org.”

[Concern #14] The code used to generate synthetic Datasets I - IV does not appear to be included in the linked GitHub repository (I looked in the benchmark models and sample data directories). That should be included to meet PLOS ONE data sharing policies, along with a script to execute the code in the benchmark models directory for all benchmarks on the synthetic data.

Response: We added the source codes for generation of simulation data as well as benchmark models at https://github.com/datax-lab/Hi-LASSO

[Concern #15] Check the capitalization of journal names and article titles in the references section. Some have unexpected lowercase letters.

Response: We have checked the journal names and article titles in the references section.

[Concern #16]

- Please cite all relevant scientific software packages used in the hi_lasso software, such as NumPy and SciPy. See https://numpy.org/citing-numpy/ and https://scipy.org/citing-scipy/ for examples. 

Response: We cited all relevant scientific software packages used in the hi_lasso software such as glmnet, NumPy, Scipy at https://hi-lasso.readthedocs.io

[Concern #17] Line 116 of the paper: Rather than describing both "parallel" and "n_jobs", just let "n_jobs" default to 1 (the serial case). Then only one parameter is needed, and "parallel" can be removed. A special value of "n_jobs is None" or "n_jobs == 0" could use the number of CPU cores returned by "multiprocessing.cpu_count()" for automatic parallelization across all available cores.

Response: We appreciate the constructive comment. We removed the ‘parallel’ parameter and corrected the ‘n_jobs’ parameter. You can also check the code at: 

https://github.com/datax-lab/Hi-LASSO/blob/master/hi_lasso/hi_lasso.py

[Concern #18] The choice of the MIT license is good for future works to build on this one!

Response: We apologize for the confusion. We have already chosen the MIT license. You can check at: https://github.com/datax-lab/Hi-LASSO/blob/master/LICENSE.

[Concern #19] Could the Spark and non-Spark libraries be combined, or make the Spark library use the base Python library as a dependency? The two code paths look fairly unrelated right now.

Response: We apologize for the confusion. The Spark engine provides essential components and libraries to handle distributed data, and our Spark version is implemented on the Spark engine. Thus, the spark and python version cannot be combined. Note that the Python version improves the efficiency using parallel processing while providing statistical testing strategy, whereas the Spark version is for distributed data. However, we made the same interface for the function, so that anyone can the two libraries easily.

[Concern #20] The "simulation_data" folder on GitHub could include a README that indicates where the data came from or how it was generated.

Response: We added a README for how simulation data is generated. Please check: 

https://github.com/datax-lab/Hi-LASSO/blob/master/simulation_data/README.md

[Concern #21] It is my hope that the authors will consider adapting this algorithm for inclusion in a popular toolkit such as scikit-learn after publication. It seems like a helpful algorithm.

Response: We thank the reviewer for an excellent suggestion. We will try it in a near future.

---

## [Decision Letter · Decision Letter 1]

21 Nov 2022

Hi-LASSO: High-performance Python and Apache spark packages for feature selection with high-dimensional data

PONE-D-22-19015R1

Dear Dr. Kang,

We’re pleased to inform you that your manuscript has been judged scientifically suitable for publication and will be formally accepted for publication once it meets all outstanding technical requirements.

Kind regards,

Sathishkumar V E

Academic Editor

PLOS ONE

Additional Editor Comments (optional):

Reviewers' comments:

Reviewer's Responses to Questions

**Comments to the Author**

1. If the authors have adequately addressed your comments raised in a previous round of review and you feel that this manuscript is now acceptable for publication, you may indicate that here to bypass the “Comments to the Author” section, enter your conflict of interest statement in the “Confidential to Editor” section, and submit your "Accept" recommendation.

Reviewer #1: All comments have been addressed

2. Is the manuscript technically sound, and do the data support the conclusions?

Reviewer #1: Yes

3. Has the statistical analysis been performed appropriately and rigorously? 

Reviewer #1: Yes

4. Have the authors made all data underlying the findings in their manuscript fully available?

Reviewer #1: Yes

5. Is the manuscript presented in an intelligible fashion and written in standard English?

Reviewer #1: Yes

6. Review Comments to the Author

Reviewer #1: The Revised paper has incorporated all the revisions as mentioned in the last review, and now the paper looks Ok in all aspects. So, the paper stands Accepted with no further revisions.

7. PLOS authors have the option to publish the peer review history of their article (what does this mean?). If published, this will include your full peer review and any attached files.

Reviewer #1: No

---

## [Editor Report · Acceptance letter]

24 Nov 2022

PONE-D-22-19015R1 

Hi-LASSO: High-performance python and apache spark packages for feature selection with high-dimensional data 

Dear Dr. Kang:

I'm pleased to inform you that your manuscript has been deemed suitable for publication in PLOS ONE. Congratulations! Your manuscript is now with our production department. 

Kind regards, 

on behalf of

Dr. Sathishkumar V E 

Academic Editor

PLOS ONE